# Exploiting Language Power for Time Series Forecasting with Exogenous Variables

## Abstract

The World Wide Web thrives on intelligent services that depend heavily on accurate time series forecasting to navigate dynamic and evolving environments. Due to the partially-observed nature of real world, exclusively focusing on the target of interest, so-called *endogenous variables*, is insufficient for accurate forecasting, especially in web systems that are susceptible to external influences. Thus, utilizing *exogenous variables* to harness external information, i.e., forecasting with exogenous variable (FEV), is imperative. Nevertheless, as the external environment is complex and ever-evolving, inadequately capturing external influences can even lead to learning spurious correlations and invalid prediction. Fortunately, recent studies have demonstrated that large language models (LLMs) exhibit exceptional recognition capabilities across open real-world systems, including a deep understanding of exogenous environments. However, it is difficult to directly apply LLMs for FEV due to challenges of task activation, exogenous knowledge extraction, and feature space alignment. In this work, we devise ExoLLM, an LLM-driven method to sufficiently utilize Exogenous variables for time series forecasting. We begin by Meta-task Instruction to activate the knowledge transfer of LLM from natural language processing to FEV. To comprehensively understand the intricate and hierarchical influences of exogenous variables, we propose Multi-grained Prompts, encompassing diverse external influences, including natural attributes, trend correlations, and period relationships between two types of variables. Additionally, a Dual TS-Text Attention is devised to bridge the feature gap between text and numeric data in LLM. Evaluation on real-world datasets demonstrates ExoLLM's superiority in exploiting exogenous information for forecasting with open-world language knowledge. Code is available at https://anonymous.4open.science/r/ExoLLM.

## CCS Concepts

• **Computing methodologies → Artificial intelligence**.

## Keywords

Time Series Forecasting, Exogenous Variables

**ACM Reference Format:**
Anonymous Author(s). 2024. Exploiting Language Power for Time Series Forecasting with Exogenous Variables. In . ACM, New York, NY, USA, 15 pages. https://doi.org/10.1145/nnnnnnn.nnnnnnn

## 1 Introduction

The World Wide Web, as a continuously and ever-changing physical system, heavily depends on the ability to forecast and respond to shifting patterns and user behaviors [11, 12, 15, 28]. Time series forecasting is essential to modern web technologies, utilizing historical data to anticipate future web patterns and trends [20, 25, 41]. Its predictive accuracy not only enhances user experience but also drives the development of intelligent web services, ranging from personalized content recommendations [27] and web economics modeling [5] to microservice log analysis [9]. These capabilities position time series forecasting as a cornerstone in creating adaptive, data-driven web platforms [23, 37].

Recently, deep models have achieved promising progress in time series forecasting [3, 17], with most of them focusing exclusively on the target of interest, known as *endogenous variables*, to make predictions [21, 26, 36, 44]. This approach often ignores the influence of *exogenous variables* from the external environment. **Exogenous variables refer to observable data within a system that are not the target variable being predicted.** As shown in Figure 1 (a), the variations within web page views (endogenous variable) are often influenced by exogenous variables, such as traffic flow, hospitalization rate, and societal events [35]. Thus, given the complex and changing physical system [48], incorporating exogenous factors, i.e., forecasting with exogenous variables (FEV) is becoming prevalent and indispensable [24].

Generally, the core of FEV is to effectively model the influence of exogenous variables on endogenous variable [4, 18, 19, 30]. Recent research in FEV proposes using attention among observed numerical exogenous series and endogenous series to capture this inherent relationship [24, 35]. Nevertheless, due to the *Intricate influences and interactions* from external environment, relying solely on time series modality is insufficient for capturing these external influences: **(1) Multi-grained temporal dependencies** [14]. The external influences and interactions from exogenous variables is multi-grained, such as periodicity and trends, which can be reflected by various aspects including complex human behaviors and living habits [46]. It is difficult to model such changing and diverse impact only by observed numeric [20], highlighting the necessity of thoroughly learning multi-grained temporal features to effectively model these intricate patterns [14]. **(2) Spurious correlation** [10]. Noise and interventions in current data can lead to learning biased external influence, thereby affecting the accuracy of forecasting results [34]. For example, traffic flows are positively correlated with exogenous weather variables, but mandatory controls can lead to less traffic even when the weather is good, resulting in spurious association that may be learned by models. Without any external knowledge from real world, a high prediction uncertainty tends to be inevitable [47].

Consequently, designing more intelligent and robust FEV framework that enable models to effectively understand the intricate

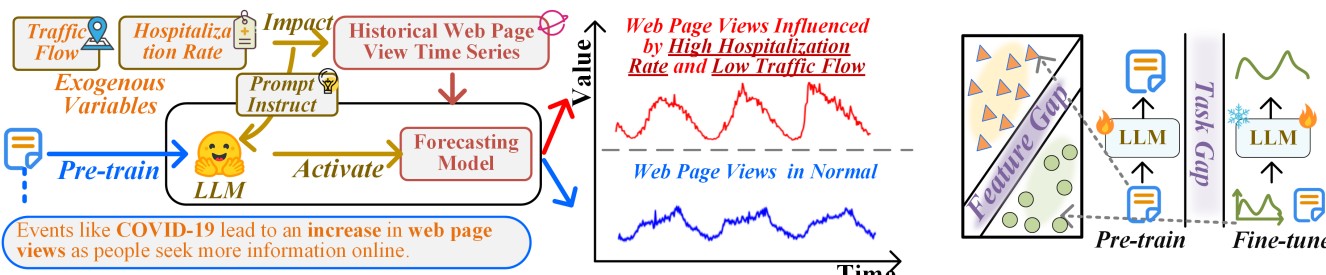

(a)LLM-empowered Forecasting with Exogenous Variables          (b) Gap in Feature and Task

**Figure 1: (a) Illustration of Knowledge Reserve from Pre-trained LLM: The extensive pre-trained text data endows LLMs with the potential to understand intricate influence of exogenous variables on web page views. (b) Huge Gaps in Feature Space and Tasks: Text embeddings and time series features are usually mapped to different feature spaces, and it is challenging to fine-tune text-generation pre-trained LLM for FEV.**

external influence and avoid spurious correlation is in demand. Fortunately, with rapid development of large language models (LLMs) [1], there have been more opportunities to leverage the vast language knowledge to comprehend external influence on endogenous variables. Through extensive training on large-scale text corpora, pre-trained LLMs have extensively acquired knowledge of multi-grained correlation between two types of variables. Intuitively, empowering FEV with these full-scale external knowledge can significantly enhance forecasting accuracy [45]. Nevertheless, as shown in Figure 1 (b), considering distinct task differences between NLP and time series forecasting [2, 45], and distant data gap between discrete text and continues numeric [14], employing LLMs to FEV faces several urgent challenges: **(1) Task activation.** How to construct task instruction to fully activate the potential of LLMs in FEV, enabling the knowledge transfer across tasks. **(2) Full-scale language-driven knowledge acquirement.** Given an LLM-based solution, how to devise effective and comprehensive prompts to acquire hierarchical and sufficient knowledge from exogenous variables. **(3) Feature space alignment.** Given the solution is concerned with two data modalities of both numerical and text data, how to construct a feasible encoding-decoding strategy to ensure the alignment between text space and time series space.

In this work, we devise ExoLLM to forecast with Exogenous variables using LLM, capturing diverse and changing external influences from exogenous variables with language-based knowledge. Technically, we elaborately craft domain-specific Meta-task Instructions to guide LLMs to process FEV tasks in different data domains. Subsequently, we establish Multi-grained Prompts to dynamically capture the natural attributes, periodic associations, trend correlations, and other granular external influence of exogenous variables, thereby adaptive transferring the dynamic auxiliary information into knowledge that can be understood by ExoLLM. Additionally, we design the Dual Time series-Text Attention Attention (DT$^2$Attention) to mitigate data discrepancies during time series encoding and feature decoding, respectively. Comprehensive evaluation demonstrates that LLM can even act as an effective few-shot and zero-shot FEV learners when adopted through our elaborate design, outperforming specialized forecasting models. Our meticulous

design enables LLMs to function even as a proficient few-shot and zero-shot FEV learner, surpassing specialized forecasting models in terms of effectiveness, as demonstrated by the comprehensive evaluation. Our contributions can be summarized as follows:

- Given the complex and evolving external environment of real-world system, i.e., web service, traffic, electricity and weather, we introduce LLMs to maximally explore the auxiliary information of exogenous variables.
- We propose ExoLLM, the first LLM-based forecasting model to accomplish FEV:
  1) To fully exploit the potential of LLM in FEV, we elaborately design Meta-task Instruction and Multi-grained Prompt, realizing the pre-trained knowledge transfer from NLP to FEV and integrate dynamic context information into knowledge of time-series domain.
  2) To deal with the distant data gap between discrete text and continues numeric, we design modality-aware encoding and decoding mechanisms, i.e., DT$^2$Attention, to achieve aligned feature before and after LLM encoding.
- ExoLLM demonstrates outstanding predictive performance across various real scenarios, including long-term, short-term, few-shot, and zero-shot forecasting. Quantitatively, ExoLLM outperforms 10 state-of-the-art models for long-term forecasting, achieving top-1 performance in 51 settings and top-2 in 5 settings out of a total of 56 settings. In addition, ExoLLM reduces MAE by an average of 4.1%, 5.2%, and 4.5% in short-term, few-shot, and zero-shot forecasting tasks, respectively.

## 2 Related Work

### 2.1 Forecasting with Exogenous Variables

In practical forecasting scenarios, the utilization of exogenous variables as auxiliary information for forecasting endogenous variables is more prevalent. Previous research has explored statistical methods such as ARIMAX [38] and SARIMAX [33], which understand relationships between exogenous and endogenous series along with auto-regression. Additionally, deep learning models like

**Table 1: Comparison between prior LLM-based time series forecasting models and ExoLLM.**

| Method | ExoLLM (Ours) | AutoTimes [2024] | TimeLLM [2023] | LLM4TS [2023] | UniTime [2024] | LLMTime [2023] | TEST [2023] | TEMPO [2023] | GPT4TS [2023] |
|---|---|---|---|---|---|---|---|---|---|
| Exogenous Variables | ✓ | ✗ | ✗ | ✗ | ✗ | ✗ | ✗ | ✗ | ✗ |
| Multimodal | ✓ | ✓ | ✓ | ✗ | ✗ | ✗ | ✓ | ✓ | ✗ |
| Feature Alignment | ✓ | ✗ | ✓ | ✗ | ✗ | ✗ | ✗ | ✗ | ✗ |

NBEATSx [29] and TiDE [6] argue that forecasting models can leverage future values of exogenous variables during the forecasing endogenous variables. Notably, TimeXer [35] introduces external information into transformer architectures through well-designed embedding strategies to effectively incorporate external information into segmented representations of endogenous variables, accommodating temporal lags or missing data records. However, these approaches rely on establishing auxiliary information only based on numeric correlation between exogenous and endogenous series. In contrast, ExoLLM has the capability to extract multi-grained effects of exogenous variables on endogenous ones as auxiliary information from extensive world knowledge, thereby holding significant potential for enhancing accuracy and generalization in FEV.

## 2.2 LLM-based Forecasting

The recent emergence of LLMs has opened up new possibilities for time series forecasting [20, 22]. GPT4TS [45] utilizes a pre-trained language model without updating its self-attention and feedforward layers. The model undergoes fine-tuning and evaluation across various time series analysis tasks, demonstrating comparable or state-of-the-art performance by leveraging knowledge transfer from natural language pre-training. LLM4TS [2] adopts a two-stage fine-tuning approach on the LLM to fully leverage time series data. TimeLLM [14] introduces the concept of text prototypes and reprograms time series based on these prototypes to align them more naturally with language models. Tempo [1] decomposes the trend, seasonality, and residual components of time series while dynamically selecting prompts to address comprehension challenges for LLMs. UniTime [20] proposes a language-empowered unified model to efficiently capture knowledge from cross-domain time series data. With their extensive knowledge base, LLMs exhibit tremendous potential in time series forecasting. However, as shown in Table 1, there has been no prior research exploiting LLM for forecasting with exogenous variables (FEV) to enhance the prediction accuracy. To address this gap, we propose ExoLLM which harnesses the power of language to capture the influence of exogenous variables on endogenous variables.

## 3 Problem Definition

In forecasting with exogenous variables, there is a historical endogenous series $\mathbf{X} \in \mathbb{R}^{1 \times L}$ and its associated exogenous information $\mathbf{E}$, where $L$ is look-back window size. Concretely, $\mathbf{E} \in \mathbb{R}^{M \times L}$ comprises multiple exogenous variables $\{\mathbf{E}^{(1)}, \mathbf{E}^{(2)}, \ldots, \mathbf{E}^{(M)}\}$, where $M$ is the variable num and $\mathbf{E}^{(m)} \in \mathbb{R}^{1 \times L}$ is the $m$-th exogenous series. Our goal is to learn a forecasting model $f(\cdot)$, which predicts the future $T$ time steps of endogenous series $\widehat{\mathbf{X}} \in \mathbb{R}^{1 \times T}$, based on its historical observation $\mathbf{X}$ and the exogenous variables $\mathbf{E}$.

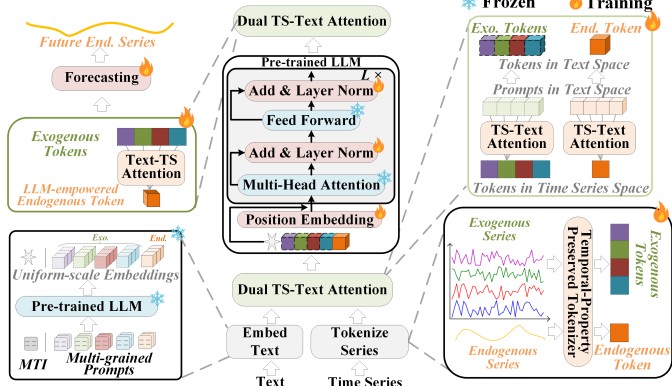

**Figure 2: Overall architecture of ExoLLM, which consists of Dual TS-Text Attention and pre-trained LLM to sufficiently exploit exogenous variables in FEV.**

## 4 Methodology

The detailed framework of ExoLLM is illustrated in Figure 2. Firstly, the Meta-task Instruction (MTI) and Multi-grained Prompt (MGP) text are embedded using frozen large language model to get uniform size embedding. Then, exogenous and endogenous series will be tokenized by shared Temporal-property preserved Tokenizer (TPT) to preserve temporal properties. Furthermore, a mainly frozen pre-trained LLM is utilized to integrate exogenous knowledge into endogenous token. It's worth noting that a Dual TS-Text Attention (DT$^2$Attention) is devised to align TS-Text feature space before and after LLM encoding, which enables the model to aware of specific modality. The output endogenous token wil be finally mapped to the future time series by a lightweight forecasting head.

### 4.1 Language-driven Exogenous Knowledge Utilization

*Meta-task Instruction.* To activate the knowledge transfer of LLM from nature language processing (NLP) to FEV, it is necessary to construct task instructions as guidance. As illustrated in Figure 3, the meta-task instruction comprises three key elements: (1) Overall description and analysis of dataset, offering explicit domain identification information to the model. (2) Brief summary of endogenous and exogenous variables, facilitating model to discern the source of each variables. (3) Introduction to the FEV task, fully activating LLM to accomplishing forecasting task with exogenous variables. We aim to activate the LLM's FEV capability in different domains through carefully designed meta-task instructions.

*Multi-grained Prompt.* To comprehensively understand the external environment of Entire systems, we need to consider not only

**[Meta-task Instruction]**
This dataset is ETTh1, containing the data collected from electricity transformers. Exogenous variables are high useful load, high useless load, middle useful load, middle useless load, low useful load, low useless load. It is necessary to utilize these exogenous variables  sequentially to predict the endogenous variable oil temperature.

**[Multi-grained Prompt]**
① **Natural attribute** ② **Trend** ③ **Periodicity** ④ **Stability**
⑤ **Noise**

**Figure 3: Example of Meta-task Instruction and headlines of Multi-grained Prompt in ETTh1.**

the apparent data correlation between numerical exogenous and endogenous variables, but also the natural properties, constant relationships, sequential trends, period influences, stability, and other multilevel factors. Therefore, we design multi-grained prompts (MGP) to exploit the LLM's comprehensive knowledge of the world to a diversified understanding of a specific environment. As shown in Table 2, the multi-grained prompt mainly consists of two elements: (1) Revealing the natural attribute of exogenous variables and their essential correlation with endogenous variables, endowing the model with prior knowledge of external environment. (2) Describing the dynamic characteristics of exogenous/endogenous series in term of trends, period, stability, and noise intensity, enabling the model to consider dynamic external influences. Intuitively, MGP not only deepens the LLM's understanding of exogenous variables, but also enhances the LLM's perception of the external invisible environment.

*Uniform-scale Text Encoding.* After constructing the meta-task instruction and multi-grained prompt, the next step involves encoding the text to obtain embeddings of uniform dimensions. To integrate these text with adequate language knowledge, we use a pre-trained LLM to encode these text descriptions. Since the text length of each prompt is different, we design an ingenious method to obtain the same embedding size. Particularly, we add a special token *<EOS>* at the end of the prompt. Since all the previous tokens are visible to *<EOS>* throughout the causal attention in LLM, the embedding of *<EOS>* could represent the entire text. The text encoding process is given by:

$$\mathbf{PT} = \text{SelectLast}(\text{LLM}(\mathbf{TD}; <EOS>)), \qquad (1)$$

where SelectLast$(\cdot)$ denotes selecting the embedding of the last *<EOS>* token, LLM$(\cdot)$ represents encoding part of large language model, $\mathbf{TD} = \{td_{\text{task}}, td_{\text{exo}}^{(1)}, td_{\text{exo}}^{(2)}, ..., td_{\text{exo}}^{(M)}, td_{\text{end}}\}$ is text description set of Meta-task Instruction and Multi-grained Prompt. $\mathbf{PT} = \{pt_{\text{task}}, pt_{\text{exo}}^{(1)}, pt_{\text{exo}}^{(2)}, ..., pt_{\text{exo}}^{(M)}, pt_{\text{end}}\}$ represents the uniform-scale text embeddings of $\mathbf{TD}$, where $pt_{\text{task}} \in \mathbb{R}^{1 \times D}$ is the embedding of meta-task instruction, $pt_{\text{exo}}^{(i)} \in \mathbb{R}^{k \times D}$ is the $i$-th exogenous prompt embedding set, $pt_{\text{end}} \in \mathbb{R}^{k \times D}$ is endogenous prompt embedding set, $k$ is multi-grained prompt number of one-type variable and $D$ is the uniform hidden dimension.

## 4.2 Temporal-property Preserved Tokenizer

To facilitate LLM's understanding of the different types of variable series, we need to compress each series into a single token. Recent

studies [21] use a linear layer to embed the entire time series as a token. However, this embedding approach neglects the temporal properties of data, resulting in the model's incomplete understanding of the relationships between exogenous and endogenous series. Therefore, we devise a Temporal-property Preserved Tokenizer (TPT) to obtain tokens reserving the temporal characteristics. Firstly, we partition the exogenous variables $\mathbf{E}$ and endogenous variables $\mathbf{X}$ into non-overlapping patches to enhance the local semantics at each time step [26], resulting in $\mathbf{P}_{end} \in \mathbb{R}^{1 \times N \times P}$ and $\mathbf{P}_{exo} \in \mathbb{R}^{M \times N \times P}$, where $P$ is patch length, and $N = \frac{L}{P}$ is the corresponding numbers of patches. To compress the temporal representations, TPT employs Self-Attention to learn temporal interactions among patches and selects the the last patch as the output:

$$\mathbf{TK}_*^{time} = \text{SelectLast}(\text{Self-Attn}(\text{PE} + \mathbf{P}_*)), \qquad (2)$$

where Self-Attn$(\cdot)$ denotes self-attention applied to time series, PE represents the position embedding, SelectLast$(\cdot)$ denotes the operation of selecting the last patch, $\mathbf{P}_*$ is patched exogenous or endogenous series and $\mathbf{TK}_*^{time}$ is the corresponding token. Selecting the last patch as the token representation of the entire series is justified by two reasons: (1) It interacts with all preceding patches through attention, thus possessing sequence-level temporal information; (2) It is closest to the future sequence, providing crucial near-term information. Finally, we obtain exogenous tokens $\mathbf{TK}_{exo}^{time} \in \mathbb{R}^{M \times D}$ and endogenous token $\mathbf{TK}_{end}^{time} \in \mathbb{R}^{1 \times D}$ in the time series feature space.

## 4.3 Knowledge-retained LLM Encoder

Understanding the exogenous impact on endogenous variables is crucial for time series forecasting. We utilize meta-task instruction along with tokenized exogenous and endogenous variables to LLMs to fully exploit the prior knowledge in LLMs, thereby forming enhanced representations of endogenous token:

$$\mathbf{TK}_{end}^{llm} = \text{LLM}(\{pt_{task}, \mathbf{TK}_{exo}, \mathbf{TK}_{end}\}), \qquad (3)$$

where LLM$(\cdot)$ denotes the encoder part of LLM. Each variable is treated as a token, and exogenous and endogenous variables are concatenated in a fixed order to form a "sentence" in a fixed order, like $[pt_{task}, \mathbf{TK}_{exo}^{(1)}, \mathbf{TK}_{exo}^{(2)}, ..., \mathbf{TK}_{exo}^{(M)}, \mathbf{TK}_{end}]$. Following [45], we freeze the positional embedding layers and self-attention blocks in LLM to retain majority of learned knowledge from language pre-training. Ultimately, we obtain an exogenous variable enhanced representation of endogenous token, $\mathbf{TK}_{end}^{llm} \in \mathbb{R}^{1 \times D}$, which encapsulates rich information from prior exogenous knowledge.

## 4.4 Feature Alignment with Dual TS-Text Attention

Given that LLM is pre-trained on discrete textual data and lack exposure to continuous numerical values, directly inputting tokens $\mathbf{TK}_{exo}^{time}$ and $\mathbf{TK}_{end}^{time}$ in time series featrue space into LLMs would increase the difficulties in understanding never-seen modality, thus resulting in degraded predictive performance. Besides, the output token from LLM in text space is difficult to decode into future series. Thus, a Dual TS-Text Attention (DT$^2$Attention) is devised to align ts-text feature space before and after LLM encoder, respectively.

**Table 2: An example of Multi-grained Prompt of one variable in ETTh1. Orange is chosen from exogenous and endogenous. Green is the variable name. Blue is prior knowledge about the variable's nature attribute. Black is the fixed template.**

| Characteristics | Prompts |
|---|---|
| Nature Attribute | ① This Exogenous variable is High UseLess Load, representing external load that is inefficiently utilized.
② Exogenous High UseLess Load indicates a potential inefficiency in the system's external load handling.
③ Exogenous High UseLess Load can lead to increased energy consumption without corresponding output.
④ Exogenous High UseLess Load might suggest that the system is operating under suboptimal external conditions. |
| Trend | ⑤ Exogenous High UseLess Load series shows an overall upward trend .
⑥ Exogenous High UseLess Load series initially rises and then declines .
⑦ Exogenous High UseLess Load series exhibits an overall declining trend .
⑧ Exogenous High UseLess Load series initially declines and then rises . |
| Period | ⑨ Exogenous High UseLess Load series has no apparent periodicity .
⑩ Exogenous High UseLess Load series exhibits shorter periodicity and higher frequency .
⑪ Exogenous High UseLess Load series displays clear periodicity .
⑫ Exogenous High UseLess Load series exhibits relatively longer periodicity . |
| Stability | ⑬ Exogenous High UseLess Load series undergoes significant instability over all the time.
⑭ Exogenous High UseLess Load series remains relatively stable with minimal fluctuations .
⑮ Exogenous High UseLess Load series experiences occasional bouts of volatility, interspersed with periods of relative calm.
⑯ Exogenous High UseLess Load series shows consistent stability, with values remaining close to a steady mean. |
| Noise Intensity | ⑰ Exogenous High UseLess Load series is subject to very strong noise interference .
⑱ Exogenous High UseLess Load series has a low signal-to-noise ratio, where noise significantly affects the clarity of the underlying data.
⑲ Exogenous High UseLess Load series experiences moderate noise, partially obscuring the underlying pattern.
⑳ Exogenous High UseLess Load series is not influenced by any noise interference . |

*TS-Text Attention.* Intuitively, there should be a certain distinction between exogenous and endogenous tokens to avoid oversmoothing representation among different types of tokens, and absorb certain prior external knowledge to enhance the LLM's encoding ability. Thus, a TS-Text Attention is designed to achieve: 1) Mapping tokens from time series feature space to text feature space; 2) Distinguishing between endogenous and exogenous Tokens. Specifically, for any type of token, TS-Text Attention designs its Query as token in time series space, while the Key and Value are its corresponding multi-grained prompt. Then, we perform Cross Attention to align tokens:

$$\mathbf{TK}_*^{text} = \text{Cross-Attn}(\mathbf{TK}_*^{time}, \mathbf{PT}_*, \mathbf{PT}_*), \tag{4}$$

where $\mathbf{TK}_*^{time} \in \mathbb{R}^D$ is the exogenous/endogenous token in time series space, $\mathbf{PT}_* \in \mathbb{R}^{k \times D}$ is this variable's corresponding multi-grained prompt, $\mathbf{TK}_*^{text} \in \mathbb{R}^D$ is the mapped token in text space and will be input into LLM in Eq (3).

*Text-TS Attention.* Denote $\mathbf{TK}_{end}^{llm} \in \mathbb{R}^{1 \times D}$ as the endogenous token encoded by LLM encoder in Eq (3). Since $\mathbf{TK}_{end}^{llm}$ remains in the text space, directly decode $\mathbf{TK}_{end}^{llm}$ for forecasting faces the challenge of converting textual semantics into time series. Thus, we use Text-TS Attention to alleviate such problem, decoding $\mathbf{TK}_{end}^{llm}$ into time series space based on the temporal information of exogenous series. This can be expressed as:

$$\mathbf{TK}_{dec}^{llm} = \text{Cross-Attn}(\mathbf{TK}_{end}^{llm}, \mathbf{TK}_{exo}^{time}, \mathbf{TK}_{exo}^{time}), \tag{5}$$

where $\mathbf{TK}_{exo}^{time}$ represents exogenous variables in time series space, $\mathbf{TK}_{dec}^{llm} \in \mathbb{R}^{1 \times D}$ is the decoded endogenous token. Through this approach, we can better utilize the representation capability of LLMs and combine exogenous series to enhance the endogenous forecasting.

## 4.5 Lightweight Forecasting Head

Considering the richness of the encoded token and maximumly preserving exogenous information by LLMs, a simple linear layer is employed to transform $\mathbf{TK}_{dec}^{llm}$ for forecasting:

$$\widehat{\mathbf{X}} = \text{Linear}(\mathbf{TK}_{dec}^{llm}). \tag{6}$$

where $\widehat{\mathbf{X}} \in \mathbb{R}^{1 \times T}$ is the future endogenous series.

## 5 Experiments
## 5.1 Dataset and Experimental Settings

To verify the model's effectiveness, we extensively evaluate our proposed ExoLLM on a diverse range of FEV scenarios, including long-term, short-term, few-shot and zero-shot task.

*Datasets and Experimental Setups.* To completely evaluate the FEV capability of ExoLLM, we conduct experiments on 12 real-world datasets. These datasets are collected from web and especially the exogenous factors retrieved from are in the formation of language. In particular, seven well-established public long-term datasets from different domains, and five short-term datasets in electricity price are involved in our FEV experiments. The endogenous and exogenous variables of each dataset are summarized in detail in Appendix A.1. For short-term forecasting datasets, the input length is set as 168 and prediction length is 24. For long-term forecasting datasets, the input length is set as 96 and prediction length varies {96, 192, 336, 720}. More implementation details can be found at Appendix A.2.

*Baselines.* We compare ExoLLM with 10 baselines, which comprise the state-of-the-art forecasting methods, including LLM-based model: LLM4TS [2], GPT4TS [45], TimeLLM [14], Transformer-based model: TimeXer [35] PatchTST [26], ITransformer [21], Crossformer [43], Autoformer [39], CNN-based model: SCINet [19], and Linear-based model: TiDE [6] . Among these models, TimeXer and

**Table 3: Full results of the long-term FEV. The input sequence length is set to 96 for all baselines. Results are averaged from all prediction lengths. The complete results are listed in the Appendix.**

| Models | ExoLLM (Ours) | | TimeXer [2024] | | ITrans. [2024] | | PatchTST [2023] | | Cross. [2023] | | TiDE [2023] | | SCINet [2022] | | Auto. 2021 | | GPT4TS [2023] | | TimeLLM [2024] | | LLM4TS [2023] | |
|---|---|---|---|---|---|---|---|---|---|---|---|---|---|---|---|---|---|---|---|---|---|---|
| Dataset | MSE | MAE | MSE | MAE | MSE | MAE | MSE | MAE | MSE | MAE | MSE | MAE | MSE | MAE | MSE | MAE | MSE | MAE | MSE | MAE | MSE | MAE |
| ECL | **0.330** | **0.404** | 0.336 | 0.415 | 0.365 | 0.486 | 0.394 | 0.446 | 0.344 | 0.412 | 0.419 | 0.468 | 0.428 | 0.450 | 0.495 | 0.528 | 0.392 | 0.442 | 0.365 | 0.413 | 0.378 | 0.427 |
| Weather | **0.001** | **0.027** | 0.002 | 0.031 | 0.002 | 0.029 | 0.002 | 0.031 | 0.005 | 0.055 | 0.002 | 0.029 | 0.007 | 0.030 | 0.007 | 0.061 | 0.005 | 0.056 | 0.003 | 0.036 | 0.004 | 0.046 |
| ETTh1 | **0.069** | **0.205** | 0.074 | 0.211 | 0.075 | 0.224 | 0.078 | 0.216 | 0.285 | 0.447 | 0.084 | 0.223 | 0.437 | 0.256 | 0.130 | 0.282 | 0.126 | 0.305 | 0.104 | 0.277 | 0.115 | 0.304 |
| ETTh2 | **0.175** | **0.327** | 0.183 | 0.337 | 0.200 | 0.357 | 0.192 | 0.345 | 1.027 | 0.873 | 0.205 | 0.356 | 1.154 | 0.406 | 0.243 | 0.386 | 0.277 | 0.443 | 0.226 | 0.388 | 0.251 | 0.415 |
| ETTm1 | **0.049** | **0.165** | 0.051 | 0.168 | 0.053 | 0.173 | 0.054 | 0.173 | 0.412 | 0.548 | 0.053 | 0.173 | 0.099 | 0.204 | 0.086 | 0.231 | 0.106 | 0.264 | 0.080 | 0.233 | 0.093 | 0.248 |
| ETTm2 | **0.113** | **0.249** | 0.116 | 0.252 | 0.127 | 0.261 | 0.120 | 0.258 | 0.976 | 0.769 | 0.122 | 0.261 | 0.685 | 0.334 | 0.154 | 0.304 | 0.196 | 0.349 | 0.162 | 0.311 | 0.179 | 0.330 |
| Traffic | **0.145** | **0.220** | 0.150 | 0.227 | 0.161 | 0.412 | 0.173 | 0.253 | 0.182 | 0.268 | 0.319 | 0.408 | 0.447 | 0.362 | 0.303 | 0.353 | 0.166 | 0.247 | 0.186 | 0.271 | 0.177 | 0.260 |

**Table 4: Full results of the short-term FEV. The input length and prediction length are set to 168 and 24 respectively for all baselines. Avg means the average results from all five datasets.**

| Models | ExoLLM (Ours) | | TimeXer [2024] | | ITrans. [2024] | | PatchTST [2023] | | Cross. [2023] | | TiDE [2023] | | SCINet [2022] | | Auto. 2021 | | GPT4TS [2023] | | TimeLLM [2024] | | LLM4TS [2023] | |
|---|---|---|---|---|---|---|---|---|---|---|---|---|---|---|---|---|---|---|---|---|---|---|
| Dataset | MSE | MAE | MSE | MAE | MSE | MAE | MSE | MAE | MSE | MAE | MSE | MAE | MSE | MAE | MSE | MAE | MSE | MAE | MSE | MAE | MSE | MAE |
| NP | **0.216** | **0.234** | 0.238 | 0.268 | 0.265 | 0.300 | 0.267 | 0.284 | 0.245 | 0.289 | 0.335 | 0.340 | 0.373 | 0.368 | 0.402 | 0.398 | 0.275 | 0.303 | 0.255 | 0.293 | 0.265 | 0.315 |
| PJM | **0.076** | **0.175** | 0.088 | 0.188 | 0.097 | 0.197 | 0.106 | 0.209 | 0.149 | 0.198 | 0.124 | 0.228 | 0.143 | 0.259 | 0.168 | 0.267 | 0.118 | 0.207 | 0.210 | 0.283 | 0.255 | 0.308 |
| BE | **0.358** | **0.225** | 0.374 | 0.241 | 0.394 | 0.270 | 0.403 | 0.264 | 0.436 | 0.294 | 0.523 | 0.336 | 0.731 | 0.412 | 0.500 | 0.333 | 0.502 | 0.288 | 0.384 | 0.230 | 0.426 | 0.258 |
| FR | **0.365** | **0.203** | 0.381 | 0.211 | 0.439 | 0.233 | 0.411 | 0.220 | 0.440 | 0.216 | 0.510 | 0.290 | 0.855 | 0.384 | 0.519 | 0.295 | 0.570 | 0.497 | 0.501 | 0.443 | 0.519 | 0.459 |
| DE | **0.422** | **0.401** | 0.440 | 0.418 | 0.479 | 0.443 | 0.461 | 0.432 | 0.540 | 0.423 | 0.568 | 0.496 | 0.565 | 0.497 | 0.674 | 0.544 | 0.569 | 0.490 | 0.498 | 0.438 | 0.517 | 0.460 |
| AVG | **0.288** | **0.251** | 0.304 | 0.265 | 0.335 | 0.289 | 0.330 | 0.282 | 0.362 | 0.284 | 0.412 | 0.338 | 0.533 | 0.384 | 0.453 | 0.368 | 0.325 | 0.326 | 0.338 | 0.378 | 0.399 | 0.408 |

TiDE are advanced recent forecaster elaborated for exogenous variables.

## 5.2 Main Results

*Long-term FEV.* Long-term forecasting results are presented in Table 3, where ExoLLM demonstrates superior performance across different prediction length again all baselines. In contrast to the cutting-edge LLM-based model TimeLLM, ExoLLM achieves performance gains of 31.2 and 19.8% in MSE and MAE metrics. Compared to the state-of-the-art (SOTA) FEV model TimeXer, ExoLLM exhibits a relative reduction of 9.1% and 4.1% in MSE and MAE metrics, respectively. These results highlight ExoLLM's exceptional FEV capability in long-term scenario.

*Short-term FEV.* In Table 4, ExoLLM consistently maintains a leading predictive performance. Compared to SOTA short-term forecasting model SCINet, ExoLLM achieves significant reductions in 35.5% MAE and 46.1% MSE respectively. Besides, ExoLLM outperforms

the FEV-designed model TimeXer in all short-term datasets. The comprehensive experimental results underscore the FEV efficacy of ExoLLM in short-term forecasting.

*Few-shot FEV.* In few-shot learning, only 10% of the training data are utilized, and the outcomes are presented in Table 5. Quantitatively, ExoLLM achieves an average 8.9% reduction in MSE and 4.5% reduction in MAE compared to the top-performing GPT4TS.

*Zero-shot FEV.* This task is to evaluate how effectively a model can perform on *target dataset* when it has been trained on *source dataset*, and the results are presented in Table 6. ExoLLM outperforms all SOTA models, achieving a performance improvement of over 5% compared to other models in zero-shot FEV. This demonstrates ExoLLM's powerful FEV generalization capabilities with pre-trained knowledge.

**Table 5: Results of few-shot FEV. Results are averaged from all prediction lengths.**

| Models | ExoLLM | | TimeXer | | ITrans. | | PatchTST | | Cross. | | TiDE | | SCINet | | Auto. | | GPT4TS | | TimeLLM | | LLM4TS | |
|---|---|---|---|---|---|---|---|---|---|---|---|---|---|---|---|---|---|---|---|---|---|---|
| Dataset | MSE | MAE | MSE | MAE | MSE | MAE | MSE | MAE | MSE | MAE | MSE | MAE | MSE | MAE | MSE | MAE | MSE | MAE | MSE | MAE | MSE | MAE |
| ETTh1 | **0.084** | **0.230** | 0.094 | 0.248 | _0.091_ | 0.251 | 0.153 | 0.344 | 0.346 | 0.506 | 0.126 | 0.312 | 0.533 | 0.288 | 0.159 | 0.316 | 0.095 | _0.242_ | 0.101 | 0.251 | 0.140 | 0.342 |
| ETTh2 | **0.253** | **0.403** | 0.279 | 0.435 | 0.290 | 0.439 | 0.401 | 0.546 | 1.501 | 1.080 | 0.327 | 0.478 | 1.681 | 0.500 | 0.352 | 0.475 | _0.278_ | _0.425_ | 0.298 | 0.439 | 0.364 | 0.512 |
| ETTm1 | **0.057** | **0.181** | 0.062 | 0.194 | 0.062 | _0.190_ | 0.124 | 0.290 | 0.475 | 0.601 | 0.094 | 0.256 | 0.115 | 0.224 | 0.102 | 0.255 | 0.062 | 0.190 | _0.062_ | 0.190 | 0.109 | 0.273 |
| ETTm2 | **0.144** | **0.291** | 0.156 | 0.310 | 0.163 | 0.306 | 0.253 | 0.410 | 1.187 | 0.882 | 0.209 | 0.365 | 0.869 | 0.389 | 0.204 | 0.360 | _0.155_ | _0.301_ | 0.158 | 0.306 | 0.231 | 0.388 |

**Table 6: Results of zero-shot FEV. Results are averaged from all prediction lengths.**

| Models | | ExoLLM | | TimeXer | | ITrans. | | PatchTST | | Cross. | | TiDE | | SCINet | | Auto. | | GPT4TS | | TimeLLM | | LLM4TS | |
|---|---|---|---|---|---|---|---|---|---|---|---|---|---|---|---|---|---|---|---|---|---|---|---|
| Source | Target | MSE | MAE | MSE | MAE | MSE | MAE | MSE | MAE | MSE | MAE | MSE | MAE | MSE | MAE | MSE | MAE | MSE | MAE | MSE | MAE | MSE | MAE |
| ETTh1 | ETTh2 | **0.204** | **0.359** | 0.228 | 0.390 | _0.221_ | 0.395 | 0.380 | 0.544 | 0.875 | 0.796 | 0.308 | 0.490 | 1.309 | 0.453 | 0.384 | 0.497 | 0.232 | _0.381_ | 0.248 | 0.394 | 0.344 | 0.538 |
| ETTh2 | ETTh1 | **0.074** | **0.212** | 0.082 | 0.228 | 0.085 | 0.230 | 0.118 | 0.287 | 0.429 | 0.562 | 0.096 | 0.251 | 0.489 | 0.262 | 0.103 | 0.250 | _0.082_ | _0.223_ | 0.087 | 0.230 | 0.107 | 0.269 |
| ETTm1 | ETTm2 | **0.162** | **0.309** | 0.177 | 0.332 | 0.178 | _0.324_ | 0.353 | 0.495 | 1.348 | 1.025 | 0.267 | 0.437 | 0.328 | 0.382 | 0.299 | 0.438 | 0.178 | 0.324 | _0.176_ | 0.324 | 0.310 | 0.466 |
| ETTm2 | ETTm1 | **0.054** | **0.176** | 0.058 | 0.187 | 0.061 | 0.185 | 0.094 | 0.248 | 0.455 | 0.538 | 0.078 | 0.220 | 0.326 | 0.236 | 0.075 | 0.217 | _0.058_ | _0.182_ | 0.059 | 0.185 | 0.086 | 0.234 |

## 5.3 Efficiency Analysis

We have compared ExoLLM with other LLM-based and linear-based methods in term of running time, and the results are provided in Table 7. As demonstrated, ExoLLM significantly reduces computational costs since it does not require repetitive text encoding during training. It saves considerable computational time compared to TimeLLM and is even comparable to DLinear. We will discuss the computational efficiency of ExoLLM from theoretical perspectives: (a)The LLM used for frozen text embeddings does not participate in forward computation or backpropagation during each training iteration. For a given dataset, its MTI and MGP components are fixed, meaning their embeddings can be precomputed and stored on disk. During training, these embeddings only need to be loaded into memory, resulting in zero additional training time for this part. (b) The TPT (Time Patch Tokenization) used for encoding time-series features reduces the sequence length by a factor of $P$ (where $P$ is the patch length). This reduces the theoretical time complexity from $O(L^2)$ to $O(\frac{L}{P}^2)$. Thus, ExoLLM is designed with a strong emphasis on resource efficiency, making it more computationally economical in practice.

**Table 7: Comparisons of per-batch running time.**

| Models | DLinear | ExoLLM | GPT4TS | TimeLLM |
|---|---|---|---|---|
| ETTh2 | 7.0ms | 24.3ms | 23.9ms | 105.4ms |
| Traffic | 47.3ms | 85.8ms | 81.3ms | 586.4ms |
| Weather | 10.1ms | 25.5ms | 20.3ms | 109.3ms |

## 5.4 Ablation Study

We conduct ablation studies by removing each module from ExoLLM on six datasets. **w/o MGP** removes Multi-grained Prompt (MGP). **w/o MTI** removes Meta-task Instruction (MTI). **w/o DT$^2$A** removes Dual TS-text Attention (DT$^2$Attention) for feature space alignment. **w/o TPT** replaces Temporal-property Preserved Tokenizer (TPT), which could preserve temporal properties for each token, with a linear layerr. We analyze the results shown in Table 8. The obervations are listed as follows: Obs.1) Removing MGP and MTI results in the most significant decrease in prediction metrics, emphasizing their strong ability in activating LLM in FEV. Obs.2) DT$^2$Attention also significantly improves the model performance, demonstrating the importance of featrure space alignment. Obs.3) TST constantly promotes the forecasting accuracy, suggesting that reserving temporal properties in each token is needed.

**Table 8: Ablation of each module on ECL, Weather, ETTh1, Traffic, PJM and NP.**

| Dataset | ECL | | Weather | | ETTh1 | | Traffic | | PJM | | NP | |
|---|---|---|---|---|---|---|---|---|---|---|---|---|
| Metric | MSE | MAE | MSE | MAE | MSE | MAE | MSE | MAE | MSE | MAE | MSE | MAE |
| ExoLLM | **0.330** | **0.404** | **0.001** | **0.027** | **0.069** | **0.205** | **0.145** | **0.220** | **0.076** | **0.175** | **0.216** | **0.234** |
| w/o MGP | 0.359 | 0.413 | 0.003 | 0.037 | 0.083 | 0.239 | 0.152 | 0.227 | 0.110 | 0.185 | 0.238 | 0.281 |
| w/o MTI | 0.354 | 0.418 | 0.003 | 0.035 | 0.096 | 0.209 | 0.153 | 0.222 | 0.099 | 0.187 | 0.238 | 0.277 |
| w/o DT$^2$A | 0.348 | 0.418 | 0.002 | 0.034 | 0.074 | 0.212 | 0.157 | 0.226 | 0.090 | 0.176 | 0.235 | 0.262 |
| w/o TPT | 0.332 | 0.405 | 0.002 | 0.031 | 0.079 | 0.210 | 0.152 | 0.223 | 0.086 | 0.182 | 0.225 | 0.241 |

## 5.5 Exogenous Scale Analysis

Real-world time series often encounter challenges such as the absence of crucial exogenous data. In this section, we employ random masking to simulate these scenarios and further investigate the forecasting performance. As illustrated in Figure 4 (a) and (b), We

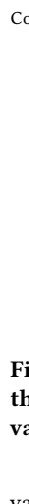
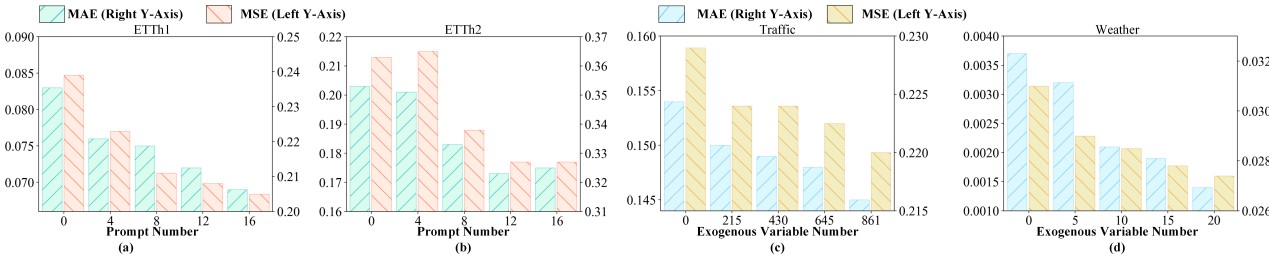

Figure 4: The MAE (left Y-axis) and MSE results (right Y-axis) of ExoLLM on ETTh, Traffic and Weather. (a) and (b) display the performance on different prompt number. (c) and (d) demonstrates the performance on different number of exogenous variables.

vary **prompt number** from 0 to 16 and report the MSE and MAE results on ETTh1 and ETTh2. For instance, when the number of prompts in Figure 4 is 16, we respectively remove prompt of Natural Attribute, Trend, Period, Stability, and Noise Intensity, and report the average results. We observe that the performance improvement is positive to the prompt size, indicating that more prompts extracting more auxiliary information from LLM. As shown in Figure 4 (c) and (d), we vary **exogenous variable number** in $\{0\%, 25\%, 50\%, 75\%, 100\%\}$ and find that more exogenous variables improve the model performance, indicating that ExoLLM is able to sufficiently understand complex and evolving environment. To further identify which prompts and exogenous variables are most important, we individually remove each exogenous prompt and variable, and results are in Table 10 and Table 9. For ETTh1, the critical exogenous prompt is Trend, and the most important exogenous variable is HUFT. This result aligns with intuition, as trend information often plays a pivotal role in long-term forecasting, and HUFT captures crucial temporal dynamics. It further demonstrates ExoLLM's ability to effectively leverage exogenous knowledge, enhancing prediction accuracy by identifying and utilizing the most relevant external factors.

**Table 9: Results of removing different type of MGP.**

|  | w/o Attribute | w/o Trend | w/o Period | w/o Stability | w/o Noise |
|---|---|---|---|---|---|
| MSE | 0.071 | 0.071 | 0.076 | 0.070 | 0.070 |
| MAE | 0.206 | 0.206 | 0.210 | 0.207 | 0.206 |

**Table 10: Results of removing different exogenous variables.**

|  | w/o HUFT | w/o HULL | w/o MUFL | w/o MULL | w/o LUFL | w/o LULL |
|---|---|---|---|---|---|---|
| MSE | 0.074 | 0.070 | 0.072 | 0.071 | 0.072 | 0.072 |
| MAE | 0.212 | 0.207 | 0.206 | 0.206 | 0.206 | 0.207 |

### 5.6 Case Study

Figure 5 illustrates the attention map of ETTh2 (comprising 6 exogenous variables and 1 endogenous variable) during causal LLM encoding. Here, tokens 0 through 7 represent the inputs to the LLM: token 0 denotes the MTI, tokens 1 through 6 represent the token sequence for the exogenous variables, and token 7 represents the

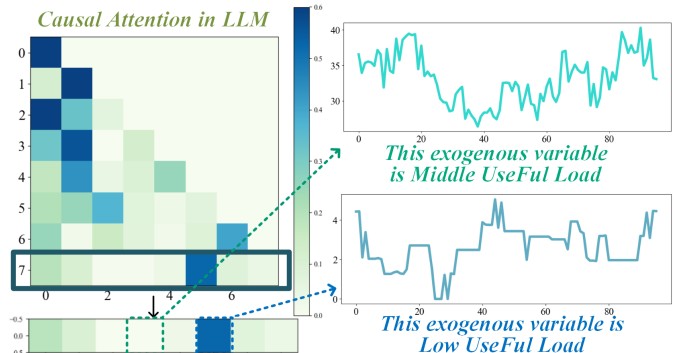

Figure 5: Case study of causal attention on different tokens.

endogenous variable. These tokens are arranged in a fixed sequence, forming an input structure akin to a sentence. Encoding through a large language model allows the endogenous variable token to be enriched with open-world knowledge conveyed through language, thereby enhancing prediction accuracy. The case study on the ETTh2 dataset demonstrates that: (1) Meta-task Instruction receive extensive attention for each variable, demonstrating that the guidance we design perfectly activates the LLM's ability to transition from NLP to FEV. (2) ExoLLM is able to distinguish between exogenous variables that exhibit strong association with the endogenous variable, resulting in a more focused and interpretable attention map.

## 6 Conclusion

To align with the evolving needs of web-related technologies, which require handling external influences from dynamic and shifting environments, we propose an LLM-based approach, ExoLLM, for time series forecasting with exogenous variables (FEV). By incorporating Meta-task Instruction, Multi-grained Prompt, and Dual TS-Text Attention, ExoLLM enables large language models (LLMs) to excel in multiple forecasting scenarios, including long-term, short-term, few-shot, and zero-shot tasks. This framework introduces a novel paradigm that taps into the textualized knowledge embedded in LLMs to enhance the understanding of structural time-series data. The versatility of ExoLLM also opens new possibilities for structural and tabular data learning across various domains, driving innovations in real-world applications central to the web ecosystem.

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

# A Experimental Details

## A.1 Dataset Descriptions

**Table 1: Dataset descriptions as summarized in [35]. *Ex.* and *En.* are abbreviations for the Exogenous variable and Endogenous variable, respectively. The dataset size is organized in (Train, Validation, Test)**

| Dataset | #Num | Ex. Descriptions | En. Descriptions | Sampling Frequency | Dataset Size |
|---|---|---|---|---|---|
| Electricity | 320 | Electricity Consumption | Electricity Consumption | 1 Hour | (18317, 2633, 5261) |
| Weather | 20 | Climate Feature | CO2-Concentration | 10 Minutes | (36792, 5271, 10540) |
| ETTh | 6 | Power Load Feature | Oil Temperature | 1 Hour | (8545, 2881, 2881) |
| ETTm | 6 | Power Load Feature | Oil Temperature | 15 Minutes | (34465, 11521, 11521) |
| Traffic | 861 | Road Occupancy Rates | Road Occupancy Rates | 1 Hour | (12185, 1757, 3509) |
| NP | 2 | Grid Load, Wind Power | Nord Pool Electricity Price | 1 Hour | (36500, 5219, 10460) |
| PJM | 2 | System Load, SyZonal COMED load | Pennsylvania-New Jersey-Maryland Electricity Price | 1 Hour | (36500, 5219, 10460) |
| BE | 2 | Generation, System Load | Belgium's Electricity Price | 1 Hour | (36500, 5219, 10460) |
| FR | 2 | Generation, System Load | France's Electricity Price | 1 Hour | (36500, 5219, 10460) |
| DE | 2 | Wind power, Ampirion zonal load | German's Electricity Price | 1 Hour | (36500, 5219, 10460) |

Seven real-world datasets are used in long-term FEV to evaluate ExoLLM, including: (1) **ETT** [44] consists of two hourly-level datasets (ETTh) and two 15minute-level datasets (ETTm). Each of them contains seven oil and load features of electricity transformers from July 2016 to July 2018. The endogenous variable is the oil temperature and the exogenous variables are 6 power load features. (2) **Weather** [44] includes 21 indicators of weather, such as air temperature, and humidity. Its data is recorded every 10 min for 2020 in Germany. In our experiment, we use the Wet Bulb factor as the endogenous variable to be predicted and the other indicators as exogenous variables. (3) **ECL** [39] contains hourly electricity consumption (in Kwh) of 321 clients from 2012 to 2014. The electricity consumption of the last client is token as an endogenous variable and other clients as exogenous variables. (4) **Traffic** [40] describes hourly road occupancy rates measured by 862 sensors on San Francisco Bay area freeways from 2015 to 2016. Te measurement of the last sensor is token as an endogenous variable and others as exogenous variables.

Five short-term datasets in term of electricity price [16] is used in short-term FEV, including: (1) **NP** represents The Nord Pool electricity market, recording the hourly electricity price, and corresponding grid load and wind power forecast from 2013-01-01 to 2018-12-24. (2) **PJM** represents the Pennsylvania-New Jersey-Maryland market, which contains the zonal electricity price in the Commonwealth Edison (COMED), and corresponding System load and COMED load forecast from 2013-01-01 to 2018-12-24. (3) **BE** represents Belgium's electricity market, recording the hourly electricity price, load forecast in Belgium, and generation forecast in France from 2011-01-09 to 2016-12-31. (4) **FR** represents the electricity market in France, recording the hourly prices, and corresponding load and generation forecast from 2012-01-09 to 2017-12-31. (5) **DE** represents the German electricity market, recording the hourly prices, the zonal load forecast in the TSO Amprion zone, and the wind and solar generation forecasts from 2012-01-09 to 2017-12-31.

## A.2 Implementation Details

All experiments are conducted using PyTorch on 2 NVIDIA H100 PCIe 80GB GPUs. We try using models such as LLaMa-7B [32], GPT-2 [7],OPT-1.3B [42] as large language models and find that

their effects are in little difference. Considering the lightweight of time series forecasting and avoid data leakage issue, GPT-2 is finally selected as the backbone of LLM in the reported results. We utilize the ADAM optimizer with L2 loss for model optimization, adjusting the batch size from 32 to 512 to maximize GPU memory utilization. Grid search is performed for learning rates, exploring values in [1e-2, 1e-3, 5e-3, 5e-4] corresponding to different datasets. Early stop of training occurs if the validation loss did not decrease for 10 consecutive rounds. The patch length is set as 8 across all datasets. The max training epoch number is set as 50. For a fair comparison, we set the input length to 96 for long-term forecasting and 168 for short-term forecasting, aligning with all baseline models.

# B Key Differences and Advantages of ExoLLM

In fact, ExoLLM is fundamentally different from other LLM-based methods. **1. Different Task Focus:** ExoLLM specifically focuses on leveraging exogenous information to enhance the predictability of the target variables. It introduces innovative components such as MGP and MTI, which significantly improve the LLM's understanding of exogenous variables, rather than merely relying on a single prompt to guide LLM. **2. Resource Efficiency:** Unlike other large language models that require LLM involvement in forward computation for text encoding. ExoLLM allows the embedding from MGP to be pre-computed and stored, significantly reducing computational time. As shown in Table 2 of the supplementary one-page PDF, ExoLLM's computational time is significantly lower, significantly shorter than other TimeLLM and competitive to other linear-based models. **3. Different Objective:** The goal of ExoLLM is to generalize across various real-world forecasting scenarios by leveraging open-world knowledge (i.e., superior insights into exogenous variables) to improve predictive performance in different contexts, a capability not shared by other LLM-based methods. **4. Appropriate Encoding:** Other LLM-based models often lack fine-grained, targeted designs for encoding different data modalities, risking suboptimal data utilization. In contrast, ExoLLM employs a tailored TFT to tokenize sequences while preserving temporal features. Additionally, the DT2Attention mechanism enables dual alignment between temporal and textual modalities during both LLM encoding and decoding, enhancing data usability—an area where other approaches fall short.

Our ExoLLM provides a paradigm to activate the power of LLM that is encompassed with textualized knowledge in open world to help better understand structural data of time-series. This can be potentially extended to more general structural and tabular data learning on various domains.

# C Necessity of Language Power

we demonstrated the importance of using LLMs for FEV from both a theoretical and practical perspective. Theoretical Justification: LLMs provide a crucial foundation for leveraging open-world knowledge to understand the impact of exogenous variables on endogenous variables. During pre-training, LLMs acquire extensive linguistic knowledge, which includes essential prior knowledge. This knowledge can be embedded into the prediction process, enhancing the model's understanding of the dynamic system. Experimental Results: In the ablation study presented in Section 5.4,

we examine the effects of removing MGP and MTI to assess the contribution of textual information to FEV prediction. The results showed that removing MGP and MTI led to the most significant decrease in prediction metrics, highlighting their critical role in activating LLMs for FEV.

## D  Data Integrity and Leakage Prevention

We consider much during the initial design of our model, particularly ensuring that ExoLLM is free from data leakage. (1)No Data Leakage in Practice: As outlined in A.2 Implementation Details, ExoLLM uses GPT-2 as its backbone. GPT-2 was trained on Web-Text, a dataset created by OpenAI using text from Reddit posts that linked to highly-rated external content prior to 2018. The datasets used in our study were all released after 2019, meaning they have no overlap with the WebText corpus. Thus, there is no potential for data leakage in principle. (2) Testing Results Confirm No Data Leakage: To further verify the absence of data leakage, we tested GPT-2 with queries about the datasets used in our study. The model

was unable to provide any relevant descriptions of these datasets, indicating that the pre-trained GPT-2 has not encountered this data before. This confirms that data leakage is not an issue. Our contribution fundamentally lies in leveraging LLMs to utilize existing knowledge about open world, enabling more practical time series forecasting. Experimental results demonstrate the effectiveness of ExoLLM on datasets where no data leakage is present.

## E  Full Results

The results presented in Tables 2, 3, and 4 highlight the predictive advantages of ExoLLM in utilizing exogenous variables. ExoLLM demonstrates strong adaptability across few-shot, long-term, and zero-shot forecasting tasks by effectively capturing multi-grained temporal dependencies. Its ability to maintain high performance even under limited (few-shot) or unseen (zero-shot) data conditions underscores the model's flexibility and generalization capabilities, making it highly applicable to dynamic, real-world scenarios.

**Table 2: Full results of the few-shot forecasting with exogenous variables.**

| Models | | ExoLLM (Ours) | | TimeXer [2024] | | ITrans. [2024] | | PatchTST [2023] | | Cross. [2023] | | TiDE [2023] | | SCINet [2022] | | Auto. 2021 | | GPT4TS [2023] | | TimeLLM [2024] | | LLM4TS [2023] | |
|---|---|---|---|---|---|---|---|---|---|---|---|---|---|---|---|---|---|---|---|---|---|---|---|
| Dataset | | MSE | MAE | MSE | MAE | MSE | MAE | MSE | MAE | MSE | MAE | MSE | MAE | MSE | MAE | MSE | MAE | MSE | MAE | MSE | MAE | MSE | MAE |
| ETTh1 | 96 | **0.068** | **0.193** | 0.076 | 0.215 | 0.075 | 0.210 | 0.111 | 0.200 | 0.174 | 0.338 | 0.116 | 0.308 | 0.449 | 0.232 | 0.156 | 0.299 | 0.072 | 0.202 | 0.077 | 0.209 | 0.113 | 0.307 |
| | 192 | **0.086** | **0.228** | 0.098 | 0.244 | 0.095 | 0.242 | 0.132 | 0.341 | 0.298 | 0.462 | 0.121 | 0.289 | 0.504 | 0.269 | 0.169 | 0.323 | 0.092 | 0.233 | 0.100 | 0.242 | 0.126 | 0.315 |
| | 336 | **0.087** | **0.232** | 0.097 | 0.250 | 0.091 | 0.250 | 0.139 | 0.339 | 0.265 | 0.440 | 0.115 | 0.279 | 0.442 | 0.272 | 0.137 | 0.289 | 0.095 | 0.240 | 0.101 | 0.250 | 0.127 | 0.309 |
| | 720 | **0.095** | **0.261** | 0.105 | 0.283 | 0.102 | 0.304 | 0.231 | 0.501 | 0.646 | 0.783 | 0.153 | 0.373 | 0.736 | 0.381 | 0.174 | 0.355 | 0.119 | 0.293 | 0.127 | 0.303 | 0.192 | 0.437 |
| | Avg. | **0.084** | **0.230** | 0.094 | 0.248 | 0.091 | 0.251 | 0.153 | 0.344 | 0.346 | 0.506 | 0.126 | 0.312 | 0.533 | 0.288 | 0.159 | 0.316 | 0.095 | 0.242 | 0.101 | 0.251 | 0.140 | 0.342 |
| ETTh2 | 96 | **0.175** | **0.328** | 0.194 | 0.352 | 0.195 | 0.345 | 0.282 | 0.457 | 0.371 | 0.498 | 0.236 | 0.388 | 1.086 | 0.402 | 0.262 | 0.404 | 0.193 | 0.344 | 0.193 | 0.344 | 0.259 | 0.423 |
| | 192 | **0.244** | **0.396** | 0.265 | 0.421 | 0.264 | 0.412 | 0.385 | 0.520 | 1.749 | 1.249 | 0.288 | 0.465 | 1.523 | 0.456 | 0.302 | 0.442 | 0.261 | 0.409 | 0.264 | 0.412 | 0.336 | 0.492 |
| | 336 | **0.275** | **0.427** | 0.308 | 0.464 | 0.310 | 0.463 | 0.440 | 0.560 | 1.367 | 1.054 | 0.359 | 0.493 | 1.626 | 0.534 | 0.377 | 0.489 | 0.304 | 0.450 | 0.324 | 0.463 | 0.400 | 0.527 |
| | 720 | **0.319** | **0.459** | 0.348 | 0.504 | 0.390 | 0.538 | 0.498 | 0.646 | 2.517 | 1.518 | 0.427 | 0.563 | 2.489 | 0.607 | 0.467 | 0.567 | 0.353 | 0.495 | 0.411 | 0.538 | 0.462 | 0.605 |
| | Avg. | **0.253** | **0.403** | 0.279 | 0.435 | 0.290 | 0.439 | 0.401 | 0.546 | 1.501 | 1.080 | 0.327 | 0.478 | 1.681 | 0.500 | 0.352 | 0.475 | 0.278 | 0.425 | 0.298 | 0.439 | 0.364 | 0.512 |
| ETTm1 | 96 | **0.034** | **0.140** | 0.037 | 0.149 | 0.038 | 0.149 | 0.072 | 0.223 | 0.224 | 0.411 | 0.058 | 0.193 | 0.065 | 0.160 | 0.127 | 0.290 | 0.038 | 0.146 | 0.039 | 0.149 | 0.065 | 0.208 |
| | 192 | **0.049** | **0.169** | 0.054 | 0.182 | 0.055 | 0.178 | 0.103 | 0.286 | 0.359 | 0.527 | 0.078 | 0.219 | 0.102 | 0.202 | 0.076 | 0.219 | 0.055 | 0.178 | 0.054 | 0.178 | 0.090 | 0.253 |
| | 336 | **0.061** | **0.191** | 0.067 | 0.205 | 0.068 | 0.199 | 0.146 | 0.309 | 0.375 | 0.543 | 0.107 | 0.288 | 0.125 | 0.240 | 0.094 | 0.248 | 0.066 | 0.199 | 0.065 | 0.199 | 0.126 | 0.299 |
| | 720 | **0.083** | **0.224** | 0.091 | 0.240 | 0.087 | 0.233 | 0.175 | 0.340 | 0.943 | 0.923 | 0.132 | 0.324 | 0.168 | 0.295 | 0.111 | 0.263 | 0.091 | 0.237 | 0.089 | 0.233 | 0.154 | 0.332 |
| | Avg. | **0.057** | **0.181** | 0.062 | 0.194 | 0.062 | 0.190 | 0.124 | 0.290 | 0.475 | 0.601 | 0.094 | 0.256 | 0.115 | 0.224 | 0.102 | 0.255 | 0.062 | 0.190 | 0.062 | 0.190 | 0.109 | 0.273 |
| ETTm2 | 96 | **0.093** | **0.234** | 0.104 | 0.250 | 0.114 | 0.263 | 0.186 | 0.376 | 0.239 | 0.409 | 0.152 | 0.291 | 0.406 | 0.303 | 0.213 | 0.373 | 0.109 | 0.249 | 0.117 | 0.263 | 0.169 | 0.334 |
| | 192 | **0.129** | **0.275** | 0.140 | 0.294 | 0.151 | 0.295 | 0.224 | 0.385 | 0.961 | 0.904 | 0.172 | 0.388 | 0.829 | 0.369 | 0.200 | 0.359 | 0.140 | 0.288 | 0.146 | 0.295 | 0.198 | 0.387 |
| | 336 | **0.154** | **0.303** | 0.167 | 0.323 | 0.175 | 0.314 | 0.282 | 0.421 | 0.684 | 0.686 | 0.241 | 0.377 | 0.973 | 0.412 | 0.195 | 0.351 | 0.160 | 0.309 | 0.164 | 0.314 | 0.261 | 0.399 |
| | 720 | **0.201** | **0.351** | 0.215 | 0.372 | 0.213 | 0.353 | 0.322 | 0.459 | 2.866 | 1.529 | 0.269 | 0.402 | 1.268 | 0.473 | 0.209 | 0.357 | 0.210 | 0.360 | 0.204 | 0.353 | 0.295 | 0.431 |
| | Avg. | **0.144** | **0.291** | 0.156 | 0.310 | 0.163 | 0.306 | 0.253 | 0.410 | 1.187 | 0.882 | 0.209 | 0.365 | 0.869 | 0.389 | 0.204 | 0.360 | 0.155 | 0.301 | 0.158 | 0.306 | 0.231 | 0.388 |

**Table 3: Full results of the long-term forecasting with exogenous variables.**

| Models | | ExoLLM (Ours) | | TimeXer [2024] | | ITrans. [2024] | | PatchTST [2023] | | Cross. [2023] | | TiDE [2023] | | SCINet [2022] | | Auto. 2021 | | GPT4TS [2023] | | TimeLLM [2024] | | LLM4TS [2023] | |
|---|---|---|---|---|---|---|---|---|---|---|---|---|---|---|---|---|---|---|---|---|---|---|---|
| Dataset | | MSE | MAE | MSE | MAE | MSE | MAE | MSE | MAE | MSE | MAE | MSE | MAE | MSE | MAE | MSE | MAE | MSE | MAE | MSE | MAE | MSE | MAE |
| ECL | 96 | 0.277 | 0.374 | 0.282 | 0.380 | 0.299 | 0.480 | 0.339 | 0.412 | 0.265 | 0.364 | 0.405 | 0.459 | 0.390 | 0.407 | 0.432 | 0.502 | 0.334 | 0.412 | 0.315 | 0.396 | 0.324 | 0.404 |
| | 192 | 0.313 | 0.389 | 0.319 | 0.399 | 0.321 | 0.461 | 0.361 | 0.425 | 0.313 | 0.390 | 0.383 | 0.442 | 0.375 | 0.433 | 0.492 | 0.492 | 0.361 | 0.420 | 0.324 | 0.378 | 0.342 | 0.399 |
| | 336 | 0.357 | 0.413 | 0.362 | 0.429 | 0.379 | 0.481 | 0.393 | 0.440 | 0.380 | 0.431 | 0.418 | 0.464 | 0.468 | 0.471 | 0.508 | 0.548 | 0.393 | 0.433 | 0.381 | 0.403 | 0.387 | 0.418 |
| | 720 | 0.372 | 0.439 | 0.380 | 0.450 | 0.461 | 0.523 | 0.482 | 0.507 | 0.418 | 0.463 | 0.471 | 0.507 | 0.477 | 0.489 | 0.547 | 0.569 | 0.481 | 0.503 | 0.439 | 0.474 | 0.460 | 0.489 |
| | AVG | 0.330 | 0.404 | 0.336 | 0.415 | 0.365 | 0.486 | 0.394 | 0.446 | 0.344 | 0.412 | 0.419 | 0.468 | 0.428 | 0.450 | 0.495 | 0.528 | 0.392 | 0.442 | 0.365 | 0.413 | 0.378 | 0.427 |
| Weather | 96 | 0.001 | 0.023 | 0.001 | 0.026 | 0.001 | 0.025 | 0.001 | 0.027 | 0.004 | 0.048 | 0.001 | 0.025 | 0.006 | 0.028 | 0.007 | 0.066 | 0.003 | 0.043 | 0.001 | 0.025 | 0.002 | 0.034 |
| | 192 | 0.001 | 0.026 | 0.002 | 0.030 | 0.002 | 0.028 | 0.002 | 0.030 | 0.005 | 0.053 | 0.001 | 0.028 | 0.007 | 0.030 | 0.007 | 0.061 | 0.004 | 0.056 | 0.003 | 0.028 | 0.004 | 0.042 |
| | 336 | 0.001 | 0.027 | 0.002 | 0.031 | 0.002 | 0.029 | 0.002 | 0.032 | 0.004 | 0.051 | 0.002 | 0.029 | 0.008 | 0.030 | 0.007 | 0.062 | 0.005 | 0.057 | 0.004 | 0.046 | 0.005 | 0.052 |
| | 720 | 0.002 | 0.031 | 0.002 | 0.035 | 0.002 | 0.033 | 0.002 | 0.036 | 0.007 | 0.067 | 0.002 | 0.033 | 0.008 | 0.033 | 0.005 | 0.053 | 0.008 | 0.066 | 0.005 | 0.044 | 0.007 | 0.055 |
| | AVG | 0.001 | 0.027 | 0.002 | 0.031 | 0.002 | 0.029 | 0.002 | 0.031 | 0.005 | 0.055 | 0.002 | 0.029 | 0.007 | 0.030 | 0.007 | 0.061 | 0.005 | 0.056 | 0.003 | 0.036 | 0.004 | 0.046 |
| ETTh1 | 96 | 0.052 | 0.176 | 0.055 | 0.180 | 0.057 | 0.185 | 0.055 | 0.178 | 0.133 | 0.297 | 0.059 | 0.184 | 0.343 | 0.204 | 0.119 | 0.263 | 0.085 | 0.170 | 0.088 | 0.271 | 0.087 | 0.270 |
| | 192 | 0.067 | 0.202 | 0.073 | 0.206 | 0.074 | 0.214 | 0.072 | 0.206 | 0.232 | 0.409 | 0.078 | 0.214 | 0.393 | 0.238 | 0.132 | 0.286 | 0.103 | 0.302 | 0.094 | 0.256 | 0.099 | 0.279 |
| | 336 | 0.080 | 0.223 | 0.085 | 0.229 | 0.084 | 0.240 | 0.087 | 0.231 | 0.244 | 0.423 | 0.093 | 0.240 | 0.406 | 0.261 | 0.126 | 0.278 | 0.128 | 0.326 | 0.106 | 0.268 | 0.117 | 0.297 |
| | 720 | 0.078 | 0.220 | 0.082 | 0.227 | 0.084 | 0.256 | 0.098 | 0.247 | 0.530 | 0.660 | 0.104 | 0.255 | 0.604 | 0.321 | 0.143 | 0.299 | 0.189 | 0.423 | 0.126 | 0.314 | 0.158 | 0.368 |
| | AVG | 0.069 | 0.205 | 0.074 | 0.211 | 0.075 | 0.224 | 0.078 | 0.216 | 0.285 | 0.447 | 0.084 | 0.223 | 0.437 | 0.256 | 0.130 | 0.282 | 0.126 | 0.305 | 0.104 | 0.277 | 0.115 | 0.304 |
| ETTh2 | 96 | 0.123 | 0.272 | 0.130 | 0.278 | 0.137 | 0.286 | 0.136 | 0.285 | 0.261 | 0.413 | 0.136 | 0.285 | 0.763 | 0.333 | 0.184 | 0.335 | 0.198 | 0.379 | 0.166 | 0.322 | 0.182 | 0.350 |
| | 192 | 0.173 | 0.326 | 0.179 | 0.330 | 0.187 | 0.339 | 0.185 | 0.337 | 1.240 | 1.028 | 0.187 | 0.339 | 1.080 | 0.375 | 0.214 | 0.364 | 0.273 | 0.428 | 0.204 | 0.383 | 0.239 | 0.405 |
| | 336 | 0.196 | 0.354 | 0.209 | 0.366 | 0.221 | 0.384 | 0.217 | 0.373 | 0.974 | 0.874 | 0.231 | 0.384 | 1.159 | 0.443 | 0.269 | 0.405 | 0.314 | 0.464 | 0.256 | 0.409 | 0.285 | 0.437 |
| | 720 | 0.207 | 0.356 | 0.215 | 0.372 | 0.253 | 0.417 | 0.229 | 0.384 | 1.633 | 1.177 | 0.267 | 0.417 | 1.615 | 0.471 | 0.303 | 0.440 | 0.323 | 0.501 | 0.277 | 0.437 | 0.300 | 0.469 |
| | AVG | 0.175 | 0.327 | 0.183 | 0.337 | 0.200 | 0.357 | 0.192 | 0.345 | 1.027 | 0.873 | 0.205 | 0.356 | 1.154 | 0.406 | 0.243 | 0.386 | 0.277 | 0.443 | 0.226 | 0.388 | 0.251 | 0.415 |
| ETTm1 | 96 | 0.026 | 0.121 | 0.027 | 0.123 | 0.029 | 0.129 | 0.029 | 0.126 | 0.171 | 0.355 | 0.030 | 0.129 | 0.050 | 0.138 | 0.097 | 0.251 | 0.055 | 0.193 | 0.044 | 0.167 | 0.050 | 0.180 |
| | 192 | 0.040 | 0.152 | 0.042 | 0.156 | 0.045 | 0.160 | 0.045 | 0.160 | 0.293 | 0.474 | 0.044 | 0.160 | 0.083 | 0.182 | 0.062 | 0.197 | 0.084 | 0.258 | 0.064 | 0.197 | 0.074 | 0.227 |
| | 336 | 0.054 | 0.177 | 0.056 | 0.181 | 0.060 | 0.184 | 0.058 | 0.184 | 0.330 | 0.503 | 0.057 | 0.184 | 0.110 | 0.222 | 0.083 | 0.230 | 0.129 | 0.286 | 0.094 | 0.267 | 0.111 | 0.277 |
| | 720 | 0.075 | 0.209 | 0.078 | 0.213 | 0.079 | 0.217 | 0.082 | 0.221 | 0.852 | 0.861 | 0.080 | 0.217 | 0.152 | 0.275 | 0.100 | 0.245 | 0.158 | 0.317 | 0.119 | 0.302 | 0.139 | 0.309 |
| | AVG | 0.049 | 0.165 | 0.051 | 0.168 | 0.053 | 0.173 | 0.054 | 0.173 | 0.412 | 0.548 | 0.053 | 0.173 | 0.099 | 0.204 | 0.086 | 0.231 | 0.106 | 0.264 | 0.080 | 0.233 | 0.093 | 0.248 |
| ETTm2 | 96 | 0.058 | 0.177 | 0.062 | 0.180 | 0.071 | 0.199 | 0.068 | 0.188 | 0.149 | 0.309 | 0.073 | 0.199 | 0.253 | 0.229 | 0.133 | 0.282 | 0.116 | 0.285 | 0.095 | 0.220 | 0.105 | 0.252 |
| | 192 | 0.092 | 0.225 | 0.095 | 0.229 | 0.108 | 0.241 | 0.100 | 0.236 | 0.686 | 0.740 | 0.104 | 0.241 | 0.592 | 0.302 | 0.143 | 0.294 | 0.160 | 0.315 | 0.123 | 0.318 | 0.141 | 0.317 |
| | 336 | 0.123 | 0.266 | 0.127 | 0.270 | 0.140 | 0.276 | 0.128 | 0.271 | 0.546 | 0.602 | 0.131 | 0.276 | 0.777 | 0.362 | 0.156 | 0.308 | 0.225 | 0.369 | 0.193 | 0.331 | 0.209 | 0.350 |
| | 720 | 0.177 | 0.327 | 0.180 | 0.330 | 0.188 | 0.329 | 0.185 | 0.335 | 2.524 | 1.424 | 0.180 | 0.329 | 1.117 | 0.441 | 0.184 | 0.333 | 0.284 | 0.428 | 0.237 | 0.374 | 0.260 | 0.401 |
| | AVG | 0.113 | 0.249 | 0.116 | 0.252 | 0.127 | 0.261 | 0.120 | 0.258 | 0.976 | 0.769 | 0.122 | 0.261 | 0.685 | 0.334 | 0.154 | 0.304 | 0.196 | 0.349 | 0.162 | 0.311 | 0.179 | 0.330 |
| Traffic | 96 | 0.143 | 0.215 | 0.145 | 0.219 | 0.156 | 0.431 | 0.176 | 0.253 | 0.154 | 0.230 | 0.350 | 0.430 | 0.371 | 0.323 | 0.290 | 0.290 | 0.169 | 0.250 | 0.164 | 0.243 | 0.167 | 0.246 |
| | 192 | 0.142 | 0.214 | 0.146 | 0.220 | 0.156 | 0.404 | 0.162 | 0.243 | 0.180 | 0.256 | 0.316 | 0.405 | 0.450 | 0.307 | 0.291 | 0.291 | 0.159 | 0.241 | 0.182 | 0.252 | 0.170 | 0.246 |
| | 336 | 0.139 | 0.212 | 0.145 | 0.224 | 0.154 | 0.399 | 0.164 | 0.248 | 0.193 | 0.289 | 0.305 | 0.398 | 0.447 | 0.309 | 0.322 | 0.416 | 0.156 | 0.242 | 0.197 | 0.287 | 0.181 | 0.264 |
| | 720 | 0.157 | 0.239 | 0.165 | 0.246 | 0.177 | 0.415 | 0.189 | 0.267 | 0.199 | 0.295 | 0.305 | 0.398 | 0.521 | 0.507 | 0.307 | 0.414 | 0.182 | 0.257 | 0.201 | 0.302 | 0.192 | 0.285 |
| | AVG | 0.145 | 0.220 | 0.150 | 0.227 | 0.161 | 0.412 | 0.173 | 0.253 | 0.182 | 0.268 | 0.319 | 0.408 | 0.447 | 0.362 | 0.303 | 0.353 | 0.166 | 0.247 | 0.186 | 0.271 | 0.177 | 0.260 |
| $1^{st}$ counts | | 51 | | 0 | | 0 | | 0 | | 2 | | 0 | | 0 | | 0 | | 1 | | 2 | | 0 | |
| $2^{nd}$ counts | | 5 | | 42 | | 7 | | 1 | | 2 | | 2 | | 0 | | 0 | | 0 | | 0 | | 0 | |

**Table 4: Full results of the zero-shot forecasting with exogenous variables.**

| Source | Target | Metric | ExoLLM (Ours) MSE | MAE | TimeXer [2024] MSE | MAE | ITrans. [2024] MSE | MAE | PatchTST [2023] MSE | MAE | Cross. [2023] MSE | MAE | TiDE [2023] MSE | MAE | SCINet [2022] MSE | MAE | Auto. 2021 MSE | MAE | GPT4TS [2023] MSE | MAE | TimeLLM [2024] MSE | MAE | LLM4TS [2023] MSE | MAE |
|---|---|---|---|---|---|---|---|---|---|---|---|---|---|---|---|---|---|---|---|---|---|---|---|---|
| ETTh1 | ETTh2 | 96 | **0.146** | **0.299** | 0.162 | 0.333 | 0.160 | 0.326 | 0.239 | 0.310 | 0.373 | 0.524 | 0.248 | 0.478 | 0.963 | 0.360 | 0.334 | 0.464 | 0.154 | 0.314 | 0.166 | 0.324 | 0.243 | 0.476 |
| | | 192 | **0.197** | **0.351** | 0.226 | 0.376 | 0.218 | 0.372 | 0.303 | 0.525 | 0.683 | 0.711 | 0.277 | 0.445 | 1.157 | 0.414 | 0.389 | 0.497 | 0.212 | 0.358 | 0.230 | 0.372 | 0.290 | 0.485 |
| | | 336 | **0.208** | **0.369** | 0.232 | 0.398 | 0.218 | 0.397 | 0.332 | 0.540 | 0.634 | 0.700 | 0.276 | 0.443 | 1.056 | 0.432 | 0.328 | 0.460 | 0.226 | 0.382 | 0.242 | 0.397 | 0.304 | 0.492 |
| | | 720 | **0.266** | **0.417** | 0.294 | 0.452 | 0.286 | 0.485 | 0.646 | 0.801 | 1.807 | 1.251 | 0.430 | 0.595 | 2.060 | 0.608 | 0.488 | 0.567 | 0.334 | 0.468 | 0.355 | 0.483 | 0.538 | 0.698 |
| | | Avg. | **0.204** | **0.359** | 0.228 | 0.390 | 0.221 | 0.395 | 0.380 | 0.544 | 0.875 | 0.796 | 0.308 | 0.490 | 1.309 | 0.453 | 0.384 | 0.497 | 0.232 | 0.381 | 0.248 | 0.394 | 0.344 | 0.538 |
| ETTh2 | ETTh1 | 96 | **0.059** | **0.186** | 0.066 | 0.200 | 0.066 | 0.196 | 0.096 | 0.259 | 0.126 | 0.283 | 0.080 | 0.220 | 0.368 | 0.228 | 0.089 | 0.229 | 0.066 | 0.195 | 0.066 | 0.195 | 0.088 | 0.240 |
| | | 192 | **0.077** | **0.215** | 0.084 | 0.229 | 0.083 | 0.224 | 0.122 | 0.282 | 0.552 | 0.678 | 0.091 | 0.253 | 0.481 | 0.247 | 0.095 | 0.240 | 0.082 | 0.222 | 0.083 | 0.224 | 0.106 | 0.267 |
| | | 336 | **0.079** | **0.219** | 0.088 | 0.238 | 0.089 | 0.238 | 0.126 | 0.287 | 0.393 | 0.541 | 0.103 | 0.253 | 0.467 | 0.274 | 0.108 | 0.251 | 0.087 | 0.231 | 0.093 | 0.238 | 0.115 | 0.270 |
| | | 720 | **0.082** | **0.226** | 0.089 | 0.248 | 0.100 | 0.265 | 0.128 | 0.318 | 0.647 | 0.747 | 0.110 | 0.277 | 0.640 | 0.299 | 0.120 | 0.279 | 0.091 | 0.244 | 0.106 | 0.265 | 0.119 | 0.298 |
| | | Avg. | **0.074** | **0.212** | 0.082 | 0.228 | 0.085 | 0.230 | 0.118 | 0.287 | 0.429 | 0.562 | 0.096 | 0.251 | 0.489 | 0.262 | 0.103 | 0.250 | 0.082 | 0.223 | 0.087 | 0.230 | 0.107 | 0.269 |
| ETTm1 | ETTm2 | 96 | **0.110** | **0.250** | 0.120 | 0.267 | 0.123 | 0.267 | 0.234 | 0.399 | 0.723 | 0.733 | 0.186 | 0.345 | 0.212 | 0.285 | 0.410 | 0.519 | 0.123 | 0.260 | 0.127 | 0.267 | 0.210 | 0.372 |
| | | 192 | **0.145** | **0.292** | 0.160 | 0.315 | 0.163 | 0.307 | 0.304 | 0.495 | 1.062 | 0.911 | 0.231 | 0.378 | 0.301 | 0.350 | 0.225 | 0.378 | 0.163 | 0.307 | 0.160 | 0.307 | 0.267 | 0.437 |
| | | 336 | **0.166** | **0.318** | 0.181 | 0.341 | 0.184 | 0.331 | 0.395 | 0.515 | 1.014 | 0.904 | 0.288 | 0.480 | 0.338 | 0.399 | 0.255 | 0.413 | 0.178 | 0.331 | 0.175 | 0.331 | 0.342 | 0.497 |
| | | 720 | **0.228** | **0.377** | 0.249 | 0.403 | 0.240 | 0.391 | 0.481 | 0.572 | 2.590 | 1.553 | 0.363 | 0.545 | 0.462 | 0.496 | 0.304 | 0.442 | 0.249 | 0.399 | 0.243 | 0.391 | 0.422 | 0.558 |
| | | Avg. | **0.162** | **0.309** | 0.177 | 0.332 | 0.178 | 0.324 | 0.353 | 0.495 | 1.348 | 1.025 | 0.267 | 0.437 | 0.328 | 0.382 | 0.299 | 0.438 | 0.178 | 0.324 | 0.176 | 0.324 | 0.310 | 0.466 |
| ETTm2 | ETTm1 | 96 | **0.032** | **0.134** | 0.036 | 0.143 | 0.039 | 0.151 | 0.064 | 0.216 | 0.082 | 0.234 | 0.052 | 0.167 | 0.139 | 0.173 | 0.073 | 0.213 | 0.037 | 0.142 | 0.040 | 0.151 | 0.058 | 0.191 |
| | | 192 | **0.045** | **0.163** | 0.049 | 0.174 | 0.053 | 0.175 | 0.078 | 0.228 | 0.336 | 0.536 | 0.060 | 0.230 | 0.290 | 0.219 | 0.070 | 0.213 | 0.049 | 0.171 | 0.051 | 0.175 | 0.069 | 0.229 |
| | | 336 | **0.059** | **0.188** | 0.064 | 0.200 | 0.067 | 0.195 | 0.107 | 0.261 | 0.261 | 0.425 | 0.092 | 0.234 | 0.371 | 0.255 | 0.074 | 0.217 | 0.061 | 0.191 | 0.063 | 0.195 | 0.100 | 0.247 |
| | | 720 | **0.080** | **0.220** | 0.086 | 0.233 | 0.085 | 0.221 | 0.128 | 0.287 | 1.144 | 0.956 | 0.107 | 0.251 | 0.506 | 0.296 | 0.083 | 0.224 | 0.084 | 0.225 | 0.082 | 0.221 | 0.118 | 0.269 |
| | | Avg. | **0.054** | **0.176** | 0.058 | 0.187 | 0.061 | 0.185 | 0.094 | 0.248 | 0.455 | 0.538 | 0.078 | 0.220 | 0.326 | 0.236 | 0.075 | 0.217 | 0.058 | 0.182 | 0.059 | 0.185 | 0.086 | 0.234 |

