# OpenReview forum: "Exploiting Language Power for Time Series Forecasting with Exogenous Variables"
_ACM.org/TheWebConf/2025/Conference — WWW 2025 Poster_

### Official Review · Reviewer_TeS9 · 2024-11-23

**Novelty:** 3
**Technical Quality:** 4

**Review:**

This study proposes a method for modeling the influence of exogenous variables on time series forecasting tasks by leveraging the comprehension and knowledge capabilities of large language models. The approach seeks to guide the integration of rich external texts and their impact on time series data through large language models. Experimental results demonstrate the effectiveness of this method in enhancing performance.
Pros：
1. The proposition of how to utilize the powerful semantic capabilities of large language models to guide the modeling of multivariate time series variate correlation is a topic worthy of in-depth exploration and research.
2. This paper is well-structured and easy to understand, with excellent logical coherence and expressive ability in its writing.

Cons:
1. This paper introduces a novel conceptual term, "exogenous variables," seemingly to indicate the presentation of a new challenge. However, the Informer paper has already proposed the task of multivariate prediction for univariate time series forecasting, and the modeling of relationships between variables in multivariate time series has long been a well-discussed topic in the field of time series research. Therefore, the author needs to further elaborate on whether the innovation of this study lies solely in the introduction of large language models.
2.  A key issue this study needs to address is how to align the vastly different modalities of text and time series data, especially when the article designs rich prompts to introduce multi-level external information influences. However, the article only proposes a simple improvement to the widely-used attention mechanism structure. The ability of this structure to fully utilize rich semantic information is questionable.

**Questions:**

1. Please refer to the review section for further explanation of the study's innovative aspects and the underlying principles of the proposed alignment structure's effectiveness.
2. The study mentions that learning solely through statistical information can lead to spurious correlations. Does this work incorporate additional prior knowledge? If so, what distinguishes this work from manual design based on prior knowledge (e.g., only using highly correlated variables as inputs to the attention mechanism)?

**Reviewer Confidence:**

3: The reviewer is confident but not certain that the evaluation is correct

**Scope:**

3: The work is somewhat relevant to the Web and to the track, and is of narrow interest to a sub-community

---

### Official Review · Reviewer_oWKh · 2024-11-25

**Novelty:** 5
**Technical Quality:** 5

**Review:**

**Summary:**

This paper proposes ExoLLM, a novel approach that leverages large language models for time series forecasting with exogenous variables. The authors introduce several innovative components including Meta-task Instruction, Multi-grained Prompts, and Dual TS-Text Attention to capture complex external influences. The experiments demonstrate promising results across various scenarios.

**Strengths:**

1 The paper is mostly well presented and easy to understand.

2 The starting point of this paper is very good. Exogenous variables can indeed serve as key auxiliary information that affects time series forecasting.

**Weaknesses:**

1 The related work section should be enriched with a more comprehensive review of LLM-based time series forecasting methods. The authors may refer to 'Large Language Models for Time Series: A Survey' for a thorough coverage of this rapidly evolving field.

2 The authors should elaborate on how LLM understands and correlates textual descriptions with numerical time series features. The semantic consistency between text prototypes and temporal patterns needs to be better explained and validated.

3 The current model lacks explicit mechanisms to handle the interdependencies among exogenous variables. Multiple exogenous factors may have synergistic or offsetting effects, which should be properly modeled and analyzed.

4 The scalability analysis would be more convincing with comprehensive experiments on datasets of varying scales, including detailed performance metrics and computational resource requirements.

5 While Table 6 shows promising zero-shot results, it would be valuable to evaluate the model's robustness on more challenging cross-domain scenarios with greater domain shifts. How does ExoLLM maintain its performance when the source and target domains are substantially different?

6 Minor: The year labels in Tables 3 and 4 are inconsistent.

**Questions:**

See Weaknesses section.

**Reviewer Confidence:**

3: The reviewer is confident but not certain that the evaluation is correct

**Scope:**

3: The work is somewhat relevant to the Web and to the track, and is of narrow interest to a sub-community

---

### Official Review · Reviewer_ZFCo · 2024-11-26

**Novelty:** 4
**Technical Quality:** 4

**Review:**

The paper proposes ExoLLM, a framework that uses large language models to improve time-series forecasting by incorporating exogenous variables. By incorporating Meta-task Instructions, Multi-grained Prompts, and Dual TS-Text Attention, ExoLLM empowers large language models to excel in diverse forecasting scenarios, including long-term, short-term, few-shot, and zero-shot tasks.

**Pros:**
1. Addresses a critical challenge in time-series forecasting: incorporating complex, multi-grained exogenous information.
2. Comprehensive experiments cover diverse scenarios, including long-term, short-term, few-shot, and zero-shot learning.
3. The structure of the article and the presentation of the figures and tables appear well-organized and clear.

**Cons:**
1. The novelty of the work primarily lies in the integration of existing mechanisms rather than in the development of fundamentally new techniques.
2. Although exogenous variables do have an impact on the prediction results, I am concerned about whether the exogenous variable information obtained from the LLM is truly beneficial for the specific dataset.
3. The lack of studies on different language model variants raises concerns about the universal applicability of the proposed method, as it may rely too heavily on a specific LLM.

**Questions:**

1. In Figure 2, how are the exogenous variables in the "tokenize series" module selected? For datasets like ETTh1, what specific variables are considered "exogenous"? Are they well-validated in terms of their causal relationship with the target variable?
2. How do you ensure that the prior knowledge encoded in LLMs is relevant to the datasets used? Is the exogenous variable information obtained during the "Embed Text" stage appropriate for the current task?
3. How the granularities in MGPs are selected and adjusted based on the characteristics of the time series data, and how does this process affect the model's understanding of temporal dynamics?
4. Does the paper explicitly describe how DT2Attention specifically addresses the modality differences between time series and text data?
This includes the conversion of time series data into the text feature space and the decoding of text feature space data back into the time series feature space.

**Reviewer Confidence:**

3: The reviewer is confident but not certain that the evaluation is correct

**Scope:**

4: The work is relevant to the Web and to the track, and is of broad interest to the community

---

### Official Review · Reviewer_V21a · 2024-11-29

**Novelty:** 5
**Technical Quality:** 5

**Review:**

This paper presents a comprehensive approach to leveraging large language models (LLMs) for time series forecasting with exogenous variables (FEV). The quality of the work is high, the authors have clearly defined the problem, proposed a novel solution (ExoLLM), and demonstrated its effectiveness through extensive experiments on multiple datasets. The authors introduce a novel approach, ExoLLM, which combines the strengths of LLMs with specialized techniques for FEV. The use of meta-task instructions, multi-grained prompts, and dual TS-Text attention is innovative and addresses several key challenges in applying LLMs to time series forecasting. The technical details are well-explained, and the paper is well-structured, making it easy to follow. However, the effectiveness of ExoLLM relies heavily on the quality of pre-trained LLMs, which may limit its applicability in scenarios where such models are not readily available. Moreover, the dataset was selected with a domain focus on electricity, and applicability to other domains was not considered. While the paper discusses some limitations, such as the need for pre-computation of embeddings, it could benefit from a more in-depth discussion on potential weaknesses and future directions.

**Questions:**

1.It is mentioned in the paper that differentiating from other LLM methods, ExoLLM employs three types of methods, Exogenous Variables, Multimodal, and Feature Alignment. How are these three types of methods expressed? It seems that these three are not described in detail in the method.

2.How do baselines (e.g. TimeXer, ITrans) handle exogenous variables and how does ExoLLM differ from them? This should be analysed specifically in the experimental section, showing why the results are better than the baselines.

3.Why does the short-term dataset only consider electricity and other areas are not analysed? How well does ExoLLM work on other domains?

**Reviewer Confidence:**

3: The reviewer is confident but not certain that the evaluation is correct

**Scope:**

3: The work is somewhat relevant to the Web and to the track, and is of narrow interest to a sub-community

---

### Official Review · Reviewer_b4N9 · 2024-12-02

**Novelty:** 4
**Technical Quality:** 5

**Review:**

This paper proposes ExoLLM, a LLM-based method for Time Series Forecasting with exogenous variables.
ExoLLM leverages knowledge power of LLM to address two main challenges in using exogenous variables: capturing multi-grained temporal dependencies and mitigating spurious correlations.
To solve these challenges, it proposed domain-speicific meta-task instructions and multi-grained prompts, to guides LLM capturing intricate and granular influence of exogenous variables.
To enhance the effectiveness of fusing different type of sources, it proposed Dual TS-Text Attention to align feature spaces of endogenous and exogenous tokens.

The experimental results exhibit ExoLLM outperforms different types of time series forecasting methods in diverse scenarios such as long-term and short term forecasting, and few-shot and zero-shot learning setting. Additionally, ablation study and exogenous scale analysis highlights careful design of the proposed modules and prompts.

Pros:
* The paper is clearly written to understand.
* The paper have comprehensive evaluations on the proposed method such as diverse scenarios (long-term/short-term/few-shot/zero-shot settings), ablation study, exogenous scale analysis, and case study.
* The proposed method achieves the best prediction results among baselinse with various types of datasets.

Cons:
* The provide source code link is invalid.
* Providing more examples of meta-task instructions and multi-grained prompts for different datasets would improve reproducibility and be helpful for the community.

**Questions:**

1. The provided source code link is invalid. Could you update it?
2. In few-shot/zero-shot FEV evaluation, does training data include both endogenous and exogenous variables? which variables are adjusted?
3. Could you have more instructions/prompts for different datasets in the appendix or source code? It would be helpful for the community.
4. Lines 489-492 in Page 5, Can you elaborate on this?: ....to avoid over-smoothing representation among different types of tokens…
5. In the section 5.3 (efficiency analysis) in page 7, While ExoLLM is resource efficient by avoiding training LLM, how efficient it is compared to baselines, e.g., DLinear(2-3x faster) and TimeLLM (4-5x slower). What accounts for these differences?

Minor things:
1. Resizing the font size of Figures 2&4&5 would be great for better readability.
2. Detailed explanation about baselines highlighting the difference would be great; it is a little bit difficult to compare baselines since there are many baselines  (e.g., distinguish them in tables 3-6)
4. Line 785 in page 7, there is a typo: “layerr” -> “layer”/

**Reviewer Confidence:**

3: The reviewer is confident but not certain that the evaluation is correct

**Scope:**

4: The work is relevant to the Web and to the track, and is of broad interest to the community